# When months matter; modelling the impact of the COVID-19 pandemic on the diagnostic pathway of Motor Neurone Disease (MND)

Ella Burchill[1]◉, Vishal Rawji[2]◉, Katy Styles[3], Siobhan Rooney[4], Patrick Stone[5], Ronan Astin[6,7,8], Nikhil Sharma[2,6]*

1 Faculty of Life Sciences and Medicine, King's College London, London, United Kingdom, 2 Department of Clinical and Movement Neurosciences, Queen Square Institute of Neurology, University College London, London, United Kingdom, 3 We Care Campaign, 4 North Ireland MND Association, 5 Marie Curie Palliative Care Research Department, Division of Psychiatry, University College London, London, United Kingdom, 6 Neuromuscular complex care centre, UCLH, London, United Kingdom, 7 National Institute of Health Research, UCLH Biomedical Research centre, UCLH, London, United Kingdom, 8 Institute for Sport, Exercise and Health, University College London, London, United Kingdom

◉ These authors contributed equally to this work.
* Nikhil.sharma@ucl.ac.uk

**Data Availability Statement:** The code is now public at: https://github.com/thesharmalab-team/plosone_covid_ref.git.

## Abstract

### Background

A diagnosis of MND takes an average 10–16 months from symptom onset. Early diagnosis is important to access supportive measures to maximise quality of life. The COVID-19 pandemic has caused significant delays in NHS pathways; the majority of GP appointments now occur online with subsequent delays in secondary care assessment. Given the rapid progression of MND, patients may be disproportionately affected resulting in late stage new presentations. We used Monte Carlo simulation to model the pre-COVID-19 diagnostic pathway and then introduced plausible COVID-19 delays.

### Methods

The diagnostic pathway was modelled using gamma distributions of time taken: 1) from symptom onset to GP presentation, 2) for specialist referral, and 3) for diagnosis reached after neurology appointment. We incorporated branches to simulate delays: when patients did not attend their GP and when the GP consultation did not result in referral. An emergency presentation was triggered when diagnostic pathway time was within 30 days of projected median survival. Total time-to-diagnosis was calculated over 100,000 iterations. The pre-COVID-19 model was estimated using published data and the Improving MND Care Survey 2019. We estimated COVID-19 delays using published statistics.

### Results

The pre-COVID model reproduced known features of the MND diagnostic pathway, with a median time to diagnosis of 399 days and predicting 5.2% of MND patients present as undiagnosed emergencies. COVID-19 resulted in diagnostic delays from 558 days when only

**Funding:** This research was supported and funded by a grant from the Reta Lila Weston Trust. NS was supported by the National Institute for Health research University College London Hospitals Biomedical Research Centre.

**Competing interests:** The authors have declared that no competing interests exist.

primary care was 25% delayed, to 915 days when both primary and secondary care were 75%. The model predicted an increase in emergency presentations ranging from 15.4%-44.5%.

## Interpretations

The model suggests the COVID-19 pandemic will result in later-stage diagnoses and more emergency presentations of undiagnosed MND. Late-stage presentations may require rapid escalation to multidisciplinary care. Proactive recognition of acute and late-stage disease with altered service provision will optimise care for people with MND.

## Introduction

Motor Neurone Disease (MND), also known as Amyotrophic Lateral Sclerosis (ALS), is a rapidly progressive neurodegenerative disorder. There are nearly 1100 people diagnosed with MND each year in the UK, with nearly half dying within 18 months of symptom onset [1]. Early diagnosis is a priority for people affected by MND and a specific focus for MND charities [2]. While there are no curative treatments available, early diagnosis facilitates supportive measures such as voice banking, respiratory and nutritional support and advance care planning. It also removes diagnostic uncertainty and allows research participation. These interventions and supportive measures focus on maximising quality of life and early access is beneficial. Meanwhile complex but time sensitive decisions, such as gastrostomy tube placement, non-invasive respiratory support and tracheostomy often require careful consideration over weeks or months; late diagnosis restricts this time.

We suspect the COVID-19 pandemic is likely to have caused significant delays in the MND diagnostic pathway across the globe–this may lead to unrecognised cases or late-stage diagnosis impacting healthcare providers including general practice, palliative care, diagnostic services and neurological teams. Here we use Monte Carlo simulation modelling to explore the possible impact of COVID-19 on the diagnostic pathway to allow services to adjust proactively if appropriate.

Diagnostic delay may be a consequence of MND's relative rarity, clinical signs mimicking other syndromes and failure to recognise the significance of symptoms [3–5]. MND has a variable presentation, with different patterns of involvement including limb, bulbar and respiratory onset [6]. Furthermore, general practitioners and healthcare professionals encounter as few as one or two presentations of early MND onset symptoms throughout their career [7].

In most health care systems, the typical MND diagnostic pathway can be slow. A recent review found typical time from symptom onset to MND diagnosis to be as long as 10–16 months, with specialist referrals, misdiagnoses (including referrals to different specialists, e.g. ENT) and unnecessary procedures identified as key factors influencing diagnostic delay [8]. This recognised lag from symptom onset to diagnosis of MND led to the UK implementation of the Red Flags checklist in 2014, designed by the Motor Neurone Disease Association (MNDA), Royal College of General Practitioners (RCGP) and specialist neurologists. These are a series of checklists of likely MND symptoms with the aim to improve time for referrals, accelerate time to diagnosis and reduce delay in accessing specialist care [9]. This diagnostic delay compromises optimal disease management and enrolment into clinical trials. We considered the impact of the COVID-19 pandemic on the UK diagnostic pathway, with the understanding that COVID has disrupted many healthcare services. The secondary effects of the

COVID-19 pandemic causing diagnostic delays and postponed access to care (as we detail in the context of MND) very likely apply to other serious conditions, including but not limited to cancer diagnoses [10, 11].

The pandemic impacted almost all aspects of medical care. There was a generalised delay in seeking a GP appointment during the 'lockdown' period. Indeed, *NHS Digital* data from April and May 2020 shows there were one third fewer appointments compared to the same months in 2019 [12], consistent with further data from *Public Health England* showing reduction in consultation rates for several conditions [13]. This may be due to reduced willingness to see the GP due to fear of COVID transmission, increased waiting times for a GP appointment and possibly reduced concern for recognising "worrying" symptoms due to the pandemic.

Delays have also occurred in secondary care. COVID-19 and associated mandatory social distancing has forced specialist practice to reshape care delivery, with a vast reduction in face-to-face clinics. This has implications for diagnostic tests and investigations for MND including physical examinations, EMG and nerve conduction studies. Considering access to a multidisciplinary MND clinic where a final diagnosis is most commonly re-affirmed, it is unclear how this has been affected by the pandemic. However, with the knowledge there has been widespread limited face-to-face appointments, varying as a result of geographical location, it is likely these services have been potentially delayed for people affected by MND.

Here we use Monte Carlo simulation modelling to explore the impact of the pandemic on the MND diagnostic pathway. We model the UK pathway as the NHS provides a comprehensive and relatively simple path to diagnosis. This is important as it may alert a range of health care providers of the problem while identifying the potential constraints. To confirm the model assumptions are valid, we first replicate the pre-pandemic UK MND diagnostic pathway ensuring that the timelines match the pre-COVID-19 literature. We then challenge the model with the known constraints imposed by COVID-19 to explore the effect on time to diagnosis and emergency acute presentations of people with undiagnosed MND. We hypothesise that the COVID-19 pandemic will result in a much later stage diagnosis that may impact the patient's journey.

Establishing diagnostic delay due to COVID-19 is important. First it may directly impact a wide range of clinical services, including General Practice, A&E and specialist neurologist services, that people affected by MND access. Second, it may allow services to anticipate late stage diagnosis and alter services appropriately. For instance, we hope that GPs may be more aware of the possibility of late stage MND presentations and the challenges that they present. It may also allow rapid development of early diagnostic pathways for MND.

## Methods

We employed Monte Carlo simulation modelling to explore the UK diagnostic pathway of 100,000 virtual patients. Monte Carlo simulation modelling relies on repeated random sampling to consider the probability of different outcomes in processes that are difficult to predict. In other words, we build the diagnostic model based on known data and then simulate the pathway for 100,000 people living with MND to explore the results.

### Model architecture

We considered the diagnostic pathway from symptom onset to diagnosis according to the typical NHS diagnostic pathway, outlined in Fig 1. This involved:

1. Time taken from symptom onset to presentation to a primary care physician or GP *(S2GP)*

2. Time taken to be seen by a neurologist after GP referral *(GP2neuro)*

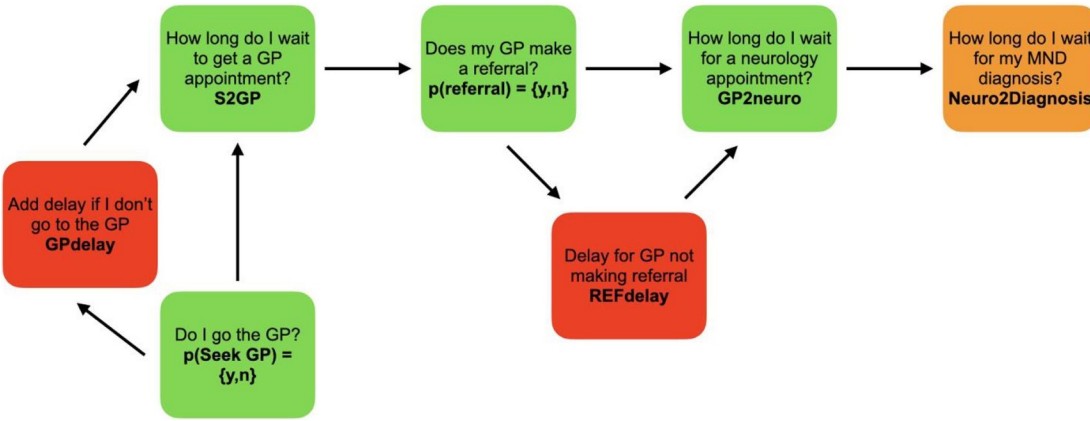

**Fig 1. Nodes illustrating the typical MND diagnostic pathway used in this model.**

3. Time taken for a diagnosis of MND to be made after being seen by a neurologist *(neuro2diagnosis)*

We included two branches to account for deviations from this serial process, both of which attempted to model real world data that accounted for diagnostic delays:

1. Delay in a patient seeking GP assessment after symptoms present *(GPdelay)*

2. A delay when a referral from GP to specialist is not made *(Refdelay)*

## Modelling the diagnostic pathway

Modelling was performed in Python 3 (pandas version 0.25.1, NumPy version 1.17.2, Seaborn version 0.12, SciPy version 1.3.1). We used gamma distributions to simulate the time taken for each step because these result in non-zero, positive integer distributions that are commonly used to model time-to-event data. One gamma distribution was produced for each step in the diagnostic pathway, each taking three inputs: shape, scale and minimum. The input: 'minimum' was the minimum time that a step could feasibly take; this served to anchor the gamma distribution at a minimum value. The shape and scale parameters were calculated using the mean and standard deviations of times taken for each process. Shape and scale were calculated using the following formulae:

$$Shape = \left(\frac{Mean}{SD}\right)^2$$

$$Scale = \frac{SD^2}{Mean}$$

We incorporated two branches in the core diagnostic pathway, that simulated deviations and diagnostic delays due to patient willingness to attend the GP and non-referrals from GP to specialist. We selected a probability (between 0–1) for each of these events occurring, denoted p1 and p2 respectively. We also formed two different gamma distributions (as above) to represent the delay if a patient did not attend their GP straight away *(GPdelay)* or if they were not referred to a specialist *(Refdelay)*. Two random numbers (r1 and r2) between 0 and 1 (one for each event) were sampled from a uniform distribution; if the random number fell above the

threshold (p1 or p2), then a sample would be drawn from the respective gamma distribution and would be added to the total time, thereby representing a delay. If the random number fell below the threshold, then no delay would be incorporated into the simulation.

Given that each step occurred in a sequential order and each step had to be completed before the next could begin, we surmised the total time to diagnosis could be calculated by summing the time taken for each process to complete. Hence, the total time to diagnosis for one simulation was given by the following:

Total time to diagnosis = S2GP + GPdelay + GP2neuro + Refdelay + neuro2diagnosis

By using Monte Carlo simulation methods, a single sample was randomly drawn from each gamma distribution of the steps shown above. These values were summed to give an estimated time from symptom onset to diagnosis, accounting for potential diagnostic delays. We repeated this process 100,000 times to simulate the diagnostic pathway for 100,000 simulated patients with MND. In doing so, we estimated the probability distribution of times-to-diagnosis for 100,000 MND patients, which captured the uncertainty in the times taken at each step, including diagnostic delays.

Given the differences in MND subtype prognosis, we hypothesised that delays due to COVID-19 would have different effects on the proportion of emergency presentations of each MND phenotype. We classified a simulation as an emergency presentation if the total time to diagnosis was within 30 days of MND subtype median time from symptom onset to death.

## Data sources and distribution estimates

We estimated S2GP times from published sources. Data from 699 patients living with MND was taken from the *2019 MND Care Survey* to inform the waiting times and distributions for the *GP2neuro* and *neuro2diagnosis* steps. In particular, we used the answers to questions 9.1 and 9.2 of the survey: *"How long did you wait between first seeing your GP about the health problems you were experiencing and first seeing a neurologist?" and "How long did you wait between first seeing a neurologist and receiving your confirmed diagnosis of MND?"*. Survey respondents could choose from one of the following options: less than 1 month, between 1 and 3 months, longer than 3 months but less than 6 months, between 6 and 9 months, longer than 9 months but less than 1 year, between 1 year and 2 years, 2 years or longer, I did not see them and can't remember. We proceeded to turn these categorised responses into a continuous distribution of times. Firstly, we omitted non-respondents and responses for *"I did not see them"* and *"can't remember"* as the times could not be determined. This resulted in 621 and 646 responses for questions 9.1 and 9.2, respectively. The categorical distributions were sampled 100,000 times to form a continuous distribution, after which they fit to a gamma distribution.

## Modelling the impact of COVID-19

We hypothesised that the COVID-19 pandemic would result in diagnostic delays by changing the probabilities of progressing through the diagnostic pathway and by changing the time taken for each step to take place. Published reports by *NHS Digital* and *Public Health England* were used to estimate the change in probability of patients seeing their GP for their initial health problems and the change in neurology referrals due to COVID-19 [12, 13]. We modelled the diagnostic delays by simulating the prolongation of primary (*S2GP* and *GPdelay*), secondary (*GP2neuro*, *Refdelay* and *neuro2diagnosis*) and all care services by 25%, 50% and 75%. As with the pre-COVID-19 model, the COVID-19 model's outputs consisted of the time for each step in the diagnostic pathway to take place, which summed to equal a total time to diagnosis. The same criterion was used to determine emergency presentations—if time to diagnosis was 30 days within time to median survival.

### Code availability

The code is available on GitHub after publication.

### Assumptions of the model

A number of assumptions were made when the diagnostic pathway was simulated and when delays due to COVID-19 were modelled, which frame the findings in this paper. Firstly, we assumed that patients progressed through the diagnostic pathway outlined above sequentially; so, patients only progressed to the next step if they had 'completed' the prior step. A second assumption was that diagnosis of MND was made at the end of this pathway, whereas in reality, diagnosis can be made at earlier stages and by different healthcare professionals other than a neurologist. Furthermore, we did not consider private consultations. Whilst these assumptions are violated in reality, the pathway simulated here represents the majority of patients with MND. As such, the findings are generalisable to the majority of patients who are eventually diagnosed with MND. Thirdly, we modelled delays in the time to diagnosis using branches in the core diagnostic pathway (*GPdelay* and *Refdelay*). Although we acknowledge that these branches are not the sole sources of diagnostic delays, we surmised that *GPdelay* and *Refdelay* would be the two delays most influential in mediating diagnostic delays due to COVID-19; this was indeed supported by published literature as being prolonged due to COVID-19. In all, incorporating these two delays maximised model relevance whilst limiting model complexity.

## Results

### Validation of the MND diagnostic pathway model

The model was able to reproduce key features of the UK MND diagnostic pathway from symptom onset to diagnosis, resulting in a median time to diagnosis of 399 days, in keeping with reported literature [8]. The model also predicted that 5.2% of patients with MND would present as emergency admissions, again consistent with previous reports [14]. This is shown in Figs 2–4.

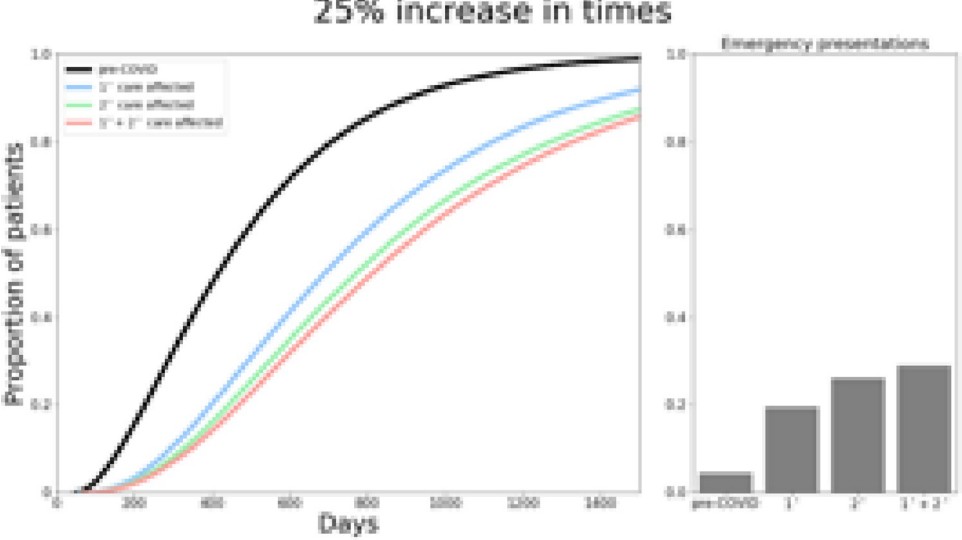

**Fig 2. Total time from symptom onset to diagnosis and proportion of emergency presentations if primary, secondary and both care services are increased by 25%.**

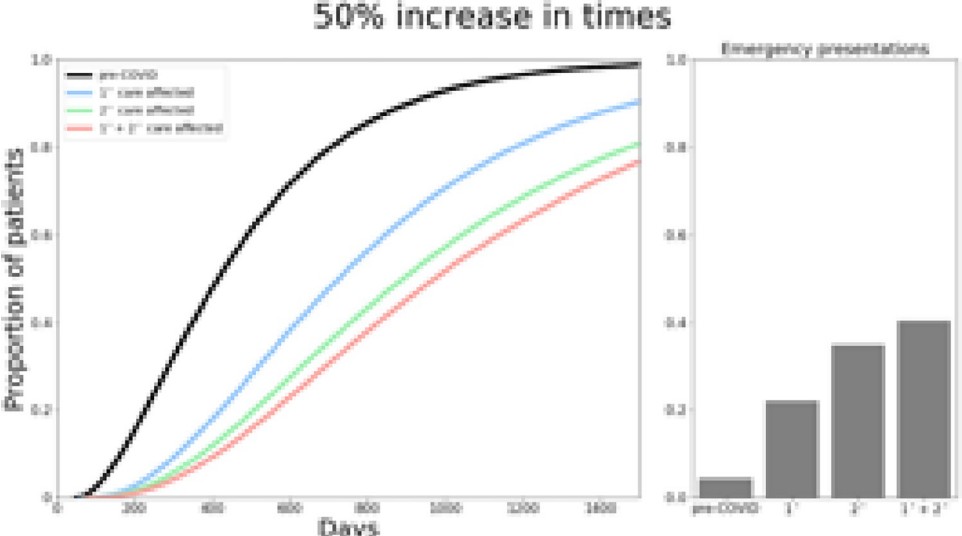

**Fig 3. Total time from symptom onset to diagnosis and proportion of emergency presentations if primary, secondary and both care services are increased by 50%.**

## Diagnostic delays due to COVID-19

We simulated diagnostic delays due to COVID-19 along two dimensions: type of care service affected by COVID-19 (primary and secondary care) and the magnitude of the delay to each step (25%, 50% and 75%). Data from NHS Digital showed that GP to neurology referrals decreased by approximately 50% since the first lockdown on 23rd March 2020. Furthermore, data from NHS England showed that approximately 48% of the public failed to seek out medical attention as they did not want to burden the NHS and out of fear of contracting COVID-19 [15]. We adjusted for these probabilities (p1 and p2) in our modelling of the COVID-19 MND diagnostic pathway.

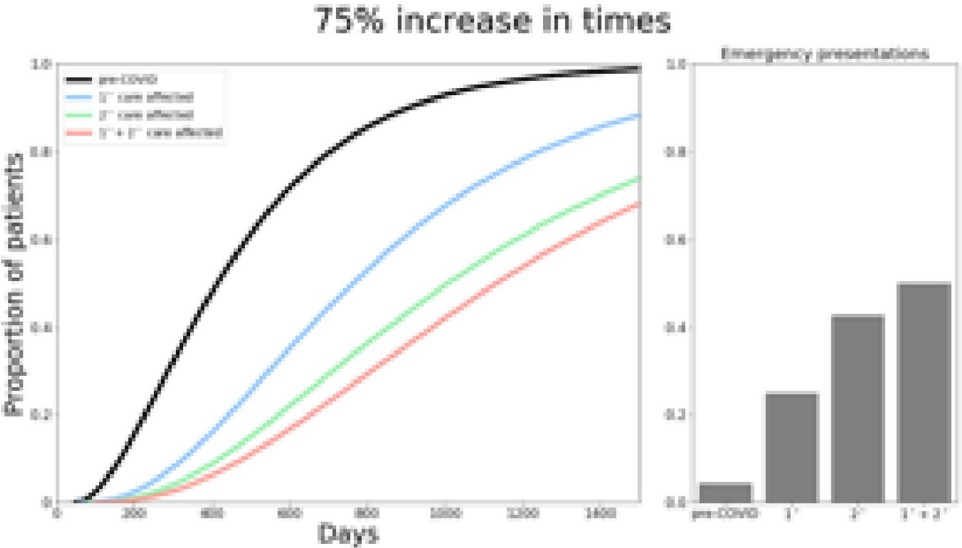

**Fig 4. Total time from symptom onset to diagnosis and proportion of emergency presentations if primary, secondary and both care services are increased by 75%.**

In the best-case scenario if time for primary care service steps were prolonged by 25%, (Fig 2), median time to diagnosis increased from 399 days to 558 days. If secondary care services were delayed by 25%, this increased to 615 days, and 656 days if both primary and secondary care services were affected by 25% delays (Fig 2). With a 50% delay (Fig 3), median total time to diagnosis would be 591, 709 and 787 days if primary, secondary or both services were impacted respectively. Finally, in the worst-case scenario, modelling an increase by 75% in service times (Fig 4) resulted in a median time to diagnosis of 626, 802 and 915 days when primary, secondary and both care services were affected.

The model predicted an increase in emergency presentations across all scenarios, ranging from 15.4%-44.5% (Fig 3). Given that the threshold for determining emergency presentations was based on median mortality from diagnosis, these results also imply the proportion of patients who die before receiving a diagnosis also increases due to COVID-19 associated delays. Furthermore, whilst 44.1% of patients were diagnosed within 12 months prior to COVID-19, this was predicted to be much lower in our simulations, ranging from 3.3%-20.1% (25%: primary 20.1%, secondary 14.9%, both 12.5%; 50%: primary 18.2%, secondary 8.9%, both 6.5%; 75%: primary 15.9%, secondary 5.4%, both 3.3%).

## Discussion

This study suggests the COVID-19 pandemic has caused significant diagnostic delays in the UK MND diagnostic pathway. The model accounts for a wide range of delays, all of which have a significant impact on the diagnostic pathway that results in more emergency admissions to A&E of people with MND who are yet to receive a diagnosis. While this model is based on the UK pathway, it is likely to apply to other similar health care systems. Delays spanning from a best to worst-case scenario will limit the quality of life for people living with MND and will undoubtedly result in a diagnosis occurring at a later stage of disease progression. This is in stark contrast to what NICE recommend as current best clinical practice: all patients with suspected MND are given a neurology appointment within four weeks of referral from the GP/ primary care provider to receive a confirmed diagnosis [16].

Importantly, the model was consistent with the literature concerning time from symptom onset to diagnosis [8]. By simulating different extents of delays, we are accounting for substantial heterogeneity in time to diagnosis and reflecting the reality of NHS service levels by illustrating a spectrum of best and worst-case scenarios. The impact of COVID-19 from an optimistic best-case scenario (with only primary care affected) was an increase in median time to diagnosis of 189 days. Compare this to a worst-case scenario where both primary and secondary care services are affected by a 75% delay, we observed a median time to diagnosis of 915 days (a delay of 516 days).

Moreover, a significant increase in emergency presentations (15.4%-44.5%) of people with undiagnosed MND was found across all scenarios of extent of delay, compared to a pre-pandemic level of 5.2%, also validated by the literature [14]. This represented acute presentations of MND without a known diagnosis, where the patient experiences rapid deterioration of symptoms and presents to the emergency department. While non-invasive ventilation is an option, setting it up in a person in whom the diagnosis is unclear may lead to suboptimal and challenging situations. These predictions have grave implications, suggesting a higher proportion of patients will die before receiving an MND diagnosis due to COVID-19 delays.

### Implications for people affected by MND

The impact of these predictions must be considered from the perspective of those living with and affected by MND. Receiving a diagnosis of a terminal condition at a later-stage in the

disease progression offers considerably less time to process and understand the implications of what this diagnosis means for the patient themselves and their family members and loved ones. This shortness of time may also limit the scope for adjustments and supportive measures offered by multidisciplinary teams to people living with MND, including voice banking, respiratory and nutritional support in addition to advance care planning. Albeit not curative, these interventions are beneficial for maximising quality of life; delaying patients' access to them can be considered to negatively impact quality of life and prognosis. There is likely to be a considerable impact on family members who may have become carers before formal diagnosis is made and support provided. Limited time also limits the scope for adequate modifications at home, requiring occupational therapy input.

There are also financial and legal implications for patients and their families. For example, if the diagnosis is given close to death, it may be not possible to access financial support and life insurance promptly. With the understanding that the diagnosis of MND carries considerable emotional and social burden, it is insensitive and unjustifiable to not give adequate time for patients and their families to make decisions about their own affairs.

## Implications for primary and acute healthcare providers

An increase in delayed diagnosis and more emergency presentations will likely impact on healthcare teams. One solution may be to update the Red Flags checklist, so primary care providers are equipped and informed to recognise different symptoms, for example shortness of breath with limb weakness may be a presenting feature of a patient with MND.

A&E services should also be prepared for the very likely increase in emergency presentations. Emergency and late-stage presentations will likely require rapid escalation to palliative care and respiratory care, services already stretched by pandemic-related increased demand. Emergency presentations of MND must be addressed in NHS Covid recovery plans. There may be pressure on the acute medical team who may need to set up non-invasive ventilation before the diagnosis is confirmed, to improve symptoms, quality of life and survival [17]. Respiratory muscle weakness may precipitate the development of daytime respiratory failure [18], an acute concern as the most common cause of death in MND [19]. In the setting of respiratory failure (the leading cause of emergency presentation) [20], this can result in tracheostomy placement, without the usual counselling, in patients who might otherwise have decided against this strategy.

## Implications for secondary healthcare providers

There are broad implications for secondary health care providers: palliative care, respiratory and gastric services and naturally, neurology and MND multidisciplinary services. Furthermore, many consultants were redeployed to assist respiratory and intensivist colleagues to help manage the acute influx of Covid-19 patients.

Our findings imply there will be increased pressure and demand for palliative care hospital services, facing increased admissions of newly diagnosed MND patients who are close to the end of life. This increase of admissions is also likely to apply to hospices community palliative care services. There is a shift towards closer collaborations between neurology and palliative care teams, for the latter to now have involvement not only in the final stages of disease procession but also to attend to the needs and concerns of the patient and family in earlier phase of the illness [21]. A palliative care approach has been shown to be helpful in improving quality of life and prognosis for MND [22, 23]. Advance care planning must be available when needed [16], this may be sooner than previously anticipated. Social and nursing end-of-life home care

services are also likely to experience increased demand if our predictions are correct; these at-home-services already face increased demands and restrictions as a result of the pandemic.

Respiratory function must be managed in people with newly diagnosed MND. Respiratory involvement of MND typically arises at a later disease stage, meaning if our model predictions are valid, more of those presenting with later-stage undiagnosed MND will require specialist respiratory care compared to the pre-pandemic MND population. This will entail more specialist respiratory investigations, including maximum respiratory pressure measurements in addition to just spirometry, since these are more sensitive to respiratory muscle weakness and have been shown to be superior predictors of hypercapnia and survival [17, 24, 25]. Later-stage diagnoses potentially poses pressure on the availability and supply of NIV equipment. Moreover, NIV is indicated in certain circumstances for COVID-19 management, with some hospital services struggling with demand already [26]. The scope for delivering NIV at home to MND patients may also be limited by the pandemic. The most frequent deterrents to NIV are cognitive impairment (consider if the patient also has frontotemporal dementia), social isolation (particularly pertinent in the pandemic), rapidly progressive disease and bulbar impairment [17].

More later-stage diagnoses of MND applies additional pressure to both community and hospital dietitians, as there may be more concerns about weight, nutrition and swallowing. From a community perspective, this could create higher demand for community feeding. BMI typically decreases with disease progression [27], a further consideration for late stage diagnoses. Gastrostomy (inserting PEG and RIG feeding tubes) will need to be discussed, which carries potential risks if unnecessarily delayed [16].

There is potential scope to manage more ALS patients remotely via the use of telemedicine tools, this has been explored in the context of the COVID-19 pandemic [28, 29]. This could involve using current technology (for example telemonitoring) that has already been trialled [30], to improve assistance, self-management and enhance quality of life. Despite the potential promise in these methods, this would have to be carefully considered on an individual patient basis, especially considering patients may be in more progressive stages of the disease.

A multidisciplinary approach to MND has proven to be most successful, considering the scope this devastating degenerative condition can have on the patient and their families. Those living with MND who have cognitive problems, such as frontotemporal dementia, need particular attention regarding their needs and wishes, as they may lack capacity and hence ability to give consent. Strengthening the links between neurology, respiratory, gastroenterology and palliative care services [31] should provide better holistic care.

We propose the implementation of rapid assessment neurology clinics for suspected MND patients, akin to the two-week wait referral pathway for suspected cancer. Another possible mitigation would be to accept referrals into the diagnostic pathway from allied health professionals, who already possess expertise in later-stage MND. This may mitigate the diagnostic delay and result in patients receiving their diagnosis sooner than our model predicts, enabling crucial access to care sooner.

## Implications for the immediate delivery of care

Our findings reflect the impact of the COVID-19 pandemic, informed by data from the first UK COVID-19 lockdown in Spring 2020, on the MND diagnostic pathway. Later-stage diagnosis and more emergency presentations must be considered an ongoing risk until access to services and population attitudes toward seeking medical attention returns to pre-pandemic states. Urgent policy is necessitated to manage the backlog within routine diagnostic services. Public health education and interventions are warranted, to ensure those who develop possible

MND symptoms are not so frightened by the pandemic that they do not seek advice from their healthcare provider. Although many secondary care services have sought to maintain access by replacing face to face consultations with telephone or video consults, it is not yet understood how well these formats perform; subtle signs and symptoms of MND more likely picked up face-to-face may be missed, leading to diagnostic delay. Other secondary care specialist investigations, including imaging, electromyography and nerve conduction studies, may also now struggle to meet the increased demand the backlog has produced. This is due to social distancing requirements and increased cleaning of equipment which reduces service efficiency. It was suggested that the under-utilised Nightingale hospitals could be revised in use for diagnostic hubs for cancer [14]. We propose this innovative approach is considered for other disease diagnostic pathways where a prompt diagnosis is pertinent and affects quality of care, MND included.

Our model accounts for the reality of the heterogeneity in NHS services by providing delays of 25%, 50% and 75%. We acknowledge our approach is an estimate of the impact of the COVID-19 pandemic on the diagnostic pathway, but by offering different extents of delay to care services, we hope to account for most uncertainty. Retrospective analysis would be valuable to appraise our modelling approach in the future.

There are limitations to our approach, we have not modelled each MND phenotype individually. We have also taken a blanket approach, not necessarily accounting for local heterogeneity in service access which may arise due to factors including ease of access to a GP, the fluctuating geographical burden of COVID-19 and related restrictions. Ease of access to a GP can be affected by the size of the GP surgery, previous history with them and personal relationship with the GP. We have not considered inequality across the healthcare system such as people from Black, Asian and Minority Ethnic (BAME) populations that typically have less access to healthcare services [32]. GP personal experience in recognising MND is also variable. Telephone consultation may amplify preconceptions in those with frailty or multimorbidity. Local services may have varying ability to secure appropriate palliative care and community AHP services; especially whilst responding to the additional posed by COVID-19.

Nevertheless, we hope our predictions can inform healthcare providers leading to interventions which will increase diagnostic capacity. By taking a proactive approach to recognition of acute and later-stage disease and altering healthcare service provision, we can optimise care and quality of life for people affected by MND, during the pandemic and in the future.

## Acknowledgments

We would like to thank the MND Association for providing data from the Improving MND Care Survey 2019.

## Author Contributions

**Conceptualization:** Ella Burchill, Vishal Rawji, Siobhan Rooney, Nikhil Sharma.

**Data curation:** Ella Burchill, Vishal Rawji.

**Formal analysis:** Vishal Rawji.

**Methodology:** Vishal Rawji, Siobhan Rooney.

**Project administration:** Ella Burchill.

**Supervision:** Nikhil Sharma.

**Writing – original draft:** Ella Burchill, Katy Styles.

**Writing – review & editing:** Ella Burchill, Katy Styles, Patrick Stone, Ronan Astin, Nikhil
Sharma.

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
