## [Decision Letter · Decision Letter 0]

3 Jun 2021

PONE-D-21-02621

When months matter; modelling the impact of the COVID-19 pandemic on the UK diagnostic pathway of Motor Neurone Disease (MND)

PLOS ONE

Thank you for submitting your manuscript to PLOS ONE. After careful consideration, we feel that it has merit but does not fully meet PLOS ONE’s publication criteria as it currently stands. Therefore, we invite you to submit a revised version of the manuscript that addresses the points raised during the review process.

We look forward to receiving your revised manuscript.

Kind regards,

Luigi Lavorgna

Academic Editor

PLOS ONE

Journal Requirements:

5. Thank you for stating the following after the Abstract Section of your manuscript:

'Funding - This research was supported and funded by a grant from the Reta Lila Weston Trust. NS was

supported by the National Institute for Health Research University College London Hospitals Biomedical

Research Centre.'

'The author(s) received no specific funding for this work. '

6. Please amend either the title on the online submission form (via Edit Submission) or the title in the manuscript so that they are identical.

7. Please amend the manuscript submission data (via Edit Submission) to include authors Vishal Rawji, Katy Styles, Siobhan Rooney, Patrick Stone, Ronan Astin and Nikhil Sharma.

Reviewers' comments:

Reviewer's Responses to Questions

**Comments to the Author**

1. Is the manuscript technically sound, and do the data support the conclusions?

Reviewer #1: Yes

Reviewer #2: Yes

2. Has the statistical analysis been performed appropriately and rigorously? 

Reviewer #1: Yes

Reviewer #2: Yes

3. Have the authors made all data underlying the findings in their manuscript fully available?

Reviewer #1: Yes

Reviewer #2: Yes

4. Is the manuscript presented in an intelligible fashion and written in standard English?

Reviewer #1: Yes

Reviewer #2: Yes

5. Review Comments to the Author

Reviewer #1: This very interesting paper explores the impact of the pandemic on the MND diagnostic pathway, suggesting that the COVID-19 pandemic will result in later-stage diagnoses and more emergency presentations of undiagnosed MND.

I would propose only minor changes:

1. Introduction section: The statement that “There are nearly 1100 people diagnosed with MND each year […]” appear to ben not clear. I would specify in which region.

2. Implications for secondary healthcare providers section: I would add in this section a small paragraph discussing the available opportunity to manage ALS patients remotely through telemedicine tools (suggested refences: Bombaci A, et al. Telemedicine for management of patients with amyotrophic lateral sclerosis through COVID-19 tail. Neurol Sci. 2021; Vasta R, et al. (2020) Telemedicine for patients with amyotrophic lateral sclerosis during COVID-19 pandemic: an Italian ALS referral center experience. Amyotrophic Lateral Sclerosis and Frontotemporal Degeneration)

Reviewer #2: In this manuscript, Burchill et al., considering that the COVID-19 pandemic has caused significant delays in NHS pathways, especially for caring disabling diseases needing multidisciplinary approaches, such as amyotrophic lateral sclerosis (ALS), and the majority of GP appointments occurred online with subsequent delays in secondary care assessment, applied a model to describe the pre-COVID-19 diagnostic pathway and then introduced plausible COVID-19 delays by using Monte Carlo simulation accounting for a wide range of delays. The pre-COVID model reproduced known features of the MND diagnostic pathway, with a median time to diagnosis of 399 days and predicting 5.2% of MND patients present as undiagnosed emergencies. COVID-19 resulted in diagnostic delays from 558 days when only primary care was 25% delayed, to 915 days when both primary and secondary care were 75%. The model predicted an increase in emergency presentations ranging from 15.4%-44.5%. The model suggests the COVID-19 pandemic will result in later-stage diagnoses and more emergency presentations of undiagnosed MND.

This is an interesting paper, focused on an innovative and crucial topic for clinical practice in the MND field. It is well written and well discussed. I have only a minor suggestion.

Among implications for secondary healthcare providers, the use of home-monitoring and services at home, such as through technological devices aimed at improving the assistance and quality of life of the monitored patients (i.e. telemonitoring respiratory function and telehealth services, such as TiM system by Hobson et al., 2018 DOI: 10.1080/21678421.2018.1440408), could be discussed in this regard, although considering the heterogeneity in healthcare services in UK, as well as in other countries.

6. PLOS authors have the option to publish the peer review history of their article (what does this mean?). If published, this will include your full peer review and any attached files.

Reviewer #1: No

Reviewer #2: No

---

## [Author Response · Author response to Decision Letter 0]

19 Oct 2021

We have responded to each of the changes suggested:

1. The formatting and referencing are now consistent with your guidelines.

2. The funding information is now updated and in the right place on the form. 

3. Our data availability statement has been amended accordingly.

4. The corresponding author is affiliated with the chosen institute, Nikhil Sharma at UCL. 

5. The titles and authors are now identical on both forms.

6. We have responded to the minor changes suggested by Reviewer 1 and 2, in the secondary care section of the discussion we have included telemedicine providers as indicated.

We hope these changes are acceptable, please get back in contact if any further adjustments to the manuscript are required.

---

## [Editor Report · Decision Letter 1]

21 Oct 2021

When months matter; modelling the impact of the COVID-19 pandemic on the diagnostic pathway of Motor Neurone Disease (MND)

PONE-D-21-02621R1

We’re pleased to inform you that your manuscript has been judged scientifically suitable for publication and will be formally accepted for publication once it meets all outstanding technical requirements.

Kind regards,

Luigi Lavorgna

Academic Editor

PLOS ONE
---

## [Editor Report · Acceptance letter]

12 May 2022

PONE-D-21-02621R1 

When months matter; modelling the impact of the COVID-19 pandemic on the diagnostic pathway of Motor Neurone Disease (MND)  

Dear Dr. Sharma:

I'm pleased to inform you that your manuscript has been deemed suitable for publication in PLOS ONE. Congratulations! Your manuscript is now with our production department. 

Kind regards, 

on behalf of

Dr. Luigi Lavorgna 

Academic Editor

PLOS ONE